# Associations of alcohol consumption and physical activity with lean type 2 diabetes mellitus among Korean adults: A prospective cohort study

Inkyung Baik[1]*, Sang Ick Park[2]

1 Department of Foods and Nutrition, College of Science and Technology, Kookmin University, Seoul, Republic of Korea, 2 Center for Biomedical Sciences, Korea National Institute of Health, Cheongju, Republic of Korea

* ibaik@kookmin.ac.kr

**Data Availability Statement:** All relevant data are within the manuscript.

**Funding:** National Research Foundation of Korea Grant funded by the Korean Government (NRF-

## Abstract

Data on the association between alcohol consumption and the risk of type 2 diabetes mellitus (T2DM) have accumulated, but little has been reported about this association in terms of lean T2DM. The present study analyzed 10-year longitudinal data to investigate the association between alcohol consumption and T2DM risk among lean individuals. This prospective study included 2,366 male and female Koreans aged 40–69 years who were free of DM and had a body mass index (BMI) <23 kg/m$^2$ during the baseline period between 2001 and 2012. Information on alcohol consumption, BMI, and incident cases of T2DM were identified by interviews and health examinations. To analyze the association between alcohol consumption and T2DM risk, Cox proportional hazard regression analysis was used. Alcohol drinkers consuming at least 16 g/day of alcohol (2 units/day) who maintained a BMI <23 kg/m$^2$ over 10 years had a significantly higher T2DM risk even after controlling for BMI and potential risk factors. Compared with lifetime abstainers, multivariate hazard ratios (HR) [95% confidence interval] of T2DM were 1.74 [1.02, 2.95] for 16–30 g/day, 2.09 [1.16, 3.77] for 31–60 g/day, and 1.94 [1.07, 3.51] for >60g/day among alcohol drinkers. No protective effect of moderate alcohol consumption <16 g/day on T2DM risk was observed. Age, parental history of DM, and physical inactivity were also significant risk factors for lean T2DM. Alcohol consumption of at least 2 units/day increased T2DM risk among lean individuals. Abstaining from alcohol and physical activity may be beneficial for the prevention of lean T2DM.

## Introduction

Epidemiological data on the association between alcohol consumption and the risk of type 2 diabetes mellitus (T2DM) have accumulated. However, dose-response results regarding this association are still inconsistent not only among individual studies but also in meta-analyses

2019R1A2C2084000) The funders had no role in study design, data collection and analysis, decision to publish, or preparation of the manuscript.

**Competing interests:** The authors have declared that no competing interests exist.

[1–3]. In particular, two recent meta-analysis studies evaluating a potential dose-response association between alcohol consumption and T2DM risk produced conflicting results among men; one found a U-shaped association indicating a protective effect of moderate alcohol consumption on T2DM risk [3], whereas the other observed an almost linear association with no reduction in T2DM risk [2]. However, both studies agreed on a U-shaped association with a protective effect of alcohol consumption on T2DM risk among women [2, 3]. Discrepancies in these sex-specific findings may be partly explained by differences in sex-associated fat distributions related to obesity. Because abdominal obesity, which is more common in men, is a strong risk factor for T2DM, the causal effect of alcohol consumption may be relatively diminished or modified by obesity among men.

T2DM patients who have a normal body mass index (BMI), which generally ranges between 18.5 kg/m$^2$ and 25 kg/m$^2$, are called 'lean T2DM' [4]. The prevalence of lean T2DM was estimated to be 6% among Caucasian men and 3% among Caucasian women who were participants in the National Health and Nutrition Examination Survey (NHANES) of the United States (US) [5]. Among Asian populations, higher prevalence estimates have been reported [5, 6]. In the Korean NHANES data collected between 2013 and 2015, the prevalence of lean T2DM was 8% among adults aged 30 years or older, and almost half of T2DM patients had a normal BMI [7]. A recent observational study using a large dataset from the German diabetes registries reported a 2.5-fold higher odds ratio of mortality in lean patients compared with obese patients [8]. However, there is little known about risk factors of T2DM morbidity and mortality among lean individuals. Earlier cross-sectional studies showed that, compared with obese diabetic patients, lean patients were more likely to be males, smokers, and alcohol drinkers [6, 8]. In particular, although there are some data from in vitro and animal model studies supporting the toxic effects of alcohol on pancreatic β-cell in terms of the pathogenesis of T2DM [9, 10], epidemiologic data showing a temporal association between alcohol consumption and the incidence of T2DM among lean individuals are limited. A lack of data on this association may be partly due to a relatively small number of cases of lean T2DM in population-based studies. Because more cases are expected in Asian populations than in non-Asian populations, it would be worthy to investigate this association in a prospective cohort study including Asian adults. Data on the associations of alcohol consumption and other risk factors with the incidence of T2DM among lean individuals may provide insights into the mortality reductions in lean diabetic patients.

The present study analyzed 10-year follow-up data from a prospective cohort study that consists of Korean adults, focusing on lean individuals, particularly those who maintained leanness over the follow-up period. We evaluated a dose-response association between alcohol consumption and T2DM risk and attempted to explore potential causative factors for lean T2DM.

## Materials and methods

### Study population

A population-based prospective study included male and female Koreans aged 40 to 69 years at baseline who were recruited as cohort participants from two cities, Ansan and Ansung. This study was initiated in 2001 to 2002 and is still ongoing as a part of the Korean Genome Epidemiology Study (KoGES). Detailed information on the study design and procedures is available elsewhere [11, 12]. Eligible cohort members who were selected via two-stage cluster sampling based on geographic and demographic characteristics were invited to visit either the Korea University Ansan Hospital or the Ajou University Medical Center for baseline health examination and questionnaire-based interview. Thus, 4,752 men and 5,261 women were initially

registered during the period of June 18, 2001 to January 29, 2003. The questionnaire captured data on socio-demographics, medical history and health conditions, family disease history, and lifestyle information, including smoking history and dietary intake, as well as alcohol consumption. Health examinations and interviews have been conducted biennially by trained researchers according to standardized protocols.

The present study analyzed data only for lean individuals with a BMI ranging from 17 kg/m$^2$ to 22.9 kg/m$^2$, which is the range for Asians used in earlier reports [6, 13]. Since the outcome of interest in this study was incident cases of T2DM, participants who were identified to have T2DM at baseline were excluded. Additionally, those who reported a diagnosis of cancer or cardiovascular disease at baseline were excluded. After making these exclusions, a total of 2338 participants (1186 men and 1152 women) were included in the analysis.

## Ethics statement

All procedures of the cohort study were performed in accordance with the Declaration of Helsinki. The study protocol was approved by the Human Subjects Review Committee either at the Korea University Ansan Hospital (IRB approval No. ED0624) or at the Ajou University Medical Center (IRB approval No. AJIRB-CRO-06-039) and written informed consent was obtained from all participants. The study datasets, which were anonymized and available for research purposes, were obtained from the Korea Centers for Disease Control and Prevention. The Human Subjects Review Committee of Kookmin University (KMU-201512-HR-094) approved the use of the datasets for this study.

## T2DM as an outcome variable

The outcome of interest in this study was incident cases of T2DM, which was defined as the use of insulin or oral glucose-lowering medication or a fasting blood glucose level ≥126 mg/dL, or a 2-hour post-load glucose level ≥200 mg/dL in a 75-g oral glucose tolerance test. Fasting and postprandial plasma specimens were collected in every visit for health examination and sent to a commercial laboratory for assays.

## Alcohol consumption as an exposure variable

Information on alcohol consumption was collected from baseline and follow-up questionnaires, which included questions about current and past alcohol consumption status and the amounts of specific alcoholic beverages consumed [12]. Participants were asked whether they had consumed alcoholic beverages at any time during their lifetimes and whether they had stopped consuming alcohol. Furthermore, they were requested to provide information on alcohol consumption during the past year, including the average frequency of drinking occasions, amount of alcohol (beer, wine, hard liquor, and traditional alcoholic drinks such as soju, chungju, and makgeolli) consumed on a typical occasion, and the volume of 1 standard drink for each alcoholic beverage. Using this information, the daily amount of alcohol consumed (g/day) was calculated and used to classify participants into five alcohol drinking groups; 0.1–5 g/day, 6–15 g/day, 16–30 g/day, 31–60 g/day, and >60 g/day. As reported in a previous study [12], the amount of alcohol consumption was strongly correlated with serum concentrations of γ-glutamyltransferase (GGT); Spearman correlation coefficient = 0.49, p-value <0.001. As non-drinking groups, lifetime abstainers and former drinkers who had abstained from alcohol consumption since the previous year were classified separately. In this study, baseline information on alcohol consumption was updated with biennial follow-up data assuming that alcohol consumption status is a changeable lifestyle factor.

## Other variables

Information on other potential risk factors including age, sex, income status, occupation, educational level, marital status, smoking status, family history of DM, waist circumference, physical activity level, and calorie intake was collected from the baseline data. Information on body weight and height, which were measured by a trained researchers in every visit for health examination, was collected from the baseline and 10-year follow-up data to calculate BMI. Detailed methodological information on anthropometric measurements and evaluations of physical activity and dietary intake is described elsewhere [12, 14].

## Statistical analysis

Descriptive statistics on the baseline characteristics of study participants were calculated according to the categories of alcohol consumption. Comparisons of characteristics across the categories were evaluated using trend tests of chi-square analysis and ANOVA. To analyze the association of a 10-year risk of T2DM with alcohol consumption and other potential risk factors, Cox proportional hazards regression analysis was used. The person-years of each participant were calculated from the date when he or she participated in the baseline examination to the date when he or she reported the first T2DM events in the follow-up examinations or until death or to December 31, 2012, whichever came first. Participants who died, or refused further participation, or were lost to follow-up were censored. The median duration of follow-up used for analysis was 8.6 years. The multivariate risk of T2DM is expressed as a hazard ratio (HR) with its 95% confidence interval (CI). In the multivariate model, age and BMI as continuous variables and sex, household income (wage $<10^6$ Won/month or $\geq 10^6$ Won/month), occupation (office worker or non-office worker), education ($<9$ years or $\geq 9$ years), marital status (married or other), parental history of DM (yes or no), smoking status (never smoked, former smoker, current smoker; $<10$ cigarettes/day, 11–20 cigarettes/day, $>20$ cigarettes/day), physical activity (quintiles of daily metabolic equivalent score), and dietary calorie intake (quintiles) as categorical variables were adjusted for. Further analyses for participants who maintained leanness through to the end of the follow-up period and sex-specific analyses were conducted. To select significant variables in the multivariate model, the stepwise method was used. All testing was based on a two-sided significance level of 0.05. SAS version 9.1.3 (SAS Institute, Cary, NC, USA) was used for all statistical analyses.

## Results

It was observed that mean daily alcohol consumption was about 20g in alcohol drinkers. In terms of the types of alcoholic beverage, a major type was soju (mean consumption: 17g of alcohol/day), which is a distilled alcoholic beverage.

Table 1 presents the descriptive statistics of the baseline characteristics for 2338 study participants across the categories of alcohol consumption.

Alcohol drinkers were more likely to be males, office workers, married, and smokers, and have a higher educational level or larger waist circumference, and engage in more activity, with increased calorie intake. They also had higher blood levels of glucose (fasting), total cholesterol, HDL-cholesterol, triglycerides, and GGT. Alcohol drinkers had lower levels of insulin (fasting) and Homeostasis Model Assessment of β-cell (HOMA-β) than lifetime abstainers. Former drinkers also showed lower HOMA-β levels and higher levels of triglycerides and GGT than lifetime abstainers.

Table 2 shows the results of the association between alcohol consumption and T2DM risk.

Among all study participants, alcohol consumption $\geq 16$ g/day increased the risk of T2DM. In particular, those who maintained leanness over the follow-up period showed stronger

**Table 1. Baseline characteristics of 2,338 study subjects across alcohol consumption groups.**

| Variables | Lifetime abstainer | Former drinker | Average alcohol consumption (g/day) | | | | | P-value for trend |
|---|---|---|---|---|---|---|---|---|
| | | | 0.1–5 | 6–15 | 16–30 | 31–60 | >60 | |
| Number of subjects (%) | 1138 (48.7) | 149 (6.4) | 382 (16.3) | 255 (10.9) | 175 (7.5) | 130 (5.6) | 109 (4.7) | |
| Age, years | 52.5±9.5 | 56.5±8.9 | 50.8±8.9 | 50.9±9.1 | 50.9±8.3 | 50.9±9.5 | 52.5±9.6 | <0.01 |
| Men, % | 25.5 | 85.2 | 46.6 | 81.2 | 89.2 | 93.9 | 97.3 | <0.001 |
| Low household income[a], % | 38.6 | 49.7 | 32.9 | 30.6 | 28.0 | 29.2 | 48.6 | 0.08 |
| Office workers, % | 4.9 | 8.7 | 8.9 | 11.4 | 9.1 | 10.8 | 9.2 | <0.001 |
| Education >9 years, % | 41.0 | 37.6 | 48.2 | 54.9 | 53.7 | 49.2 | 46.8 | <0.001 |
| Married, % | 88.2 | 91.3 | 91.1 | 93.7 | 93.1 | 95.4 | 92.7 | <0.001 |
| Current smokers, % | 14.4 | 49.7 | 26.7 | 48.6 | 55.4 | 63.1 | 67.0 | <0.001 |
| Family history of diabetes mellitus, % | 8.8 | 5.4 | 9.2 | 7.5 | 8.0 | 4.6 | 11.0 | 0.54 |
| Body mass index, kg/m$^2$ | 21.2±1.3 | 21.0±1.4 | 21.3±1.3 | 21.2±1.4 | 21.2±1.4 | 21.2±1.4 | 21.1±1.4 | 0.84 |
| Waist circumference, cm | 73.8±6.4 | 76.6±5.5 | 74.1±5.8 | 75.5±5.5 | 75.9±5.3 | 77.4±4.8 | 77.4±5.9 | <0.001 |
| Physical activity, MET-hours/d | 30.5±15.6 | 35.3±17.0 | 32.2±15.7 | 32.1±15.9 | 32.2±16.1 | 34.2±17.3 | 38.7±17.8 | <0.001 |
| Dietary calorie intake, kcal/d | 1812±616 | 1869±696 | 1856±578 | 1897±523 | 1867±477 | 1960±582 | 1935±665 | <0.05 |
| Biochemical assays | | | | | | | | |
| Glucose, mg/dL | 84.9±7.7 | 87.0±9.0 | 85.9±8.3 | 87.4±8.4 | 87.8±8.9 | 88.0±9.2 | 90.4±9.3 | <0.001 |
| Insulin, μU/mL | 6.68±4.67 | 6.42±4.09 | 6.33±3.15 | 6.06±5.11 | 5.62±2.62 | 5.36±2.70 | 5.40±2.65 | <0.001 |
| HOMA-IR | 1.41±1.05 | 1.39±0.88 | 1.35±0.71 | 1.31±1.08 | 1.23±0.62 | 1.17±0.59 | 1.22±0.63 | <0.01 |
| HOMA-β | 123.1±126.3 | 108.1±91.1 | 110.9±66.4 | 100.4±100.7 | 90.5±53.4 | 93.1±73.8 | 77.3±44.2 | <0.001 |
| Total cholesterol, mg/dL | 190.1±32.9 | 189.6±35.7 | 187.4±34.0 | 189.6±32.9 | 189.6±33.4 | 187.7±37.0 | 184.3±36.0 | 0.49 |
| HDL cholesterol, mg/dL | 52.4±11.9 | 52.6±12.9 | 53.4±12.4 | 54.7±12.4 | 56.9±14.2 | 54.4±11.9 | 59.3±15.8 | <0.001 |
| Triglycerides, mg/dL | 111.7±62.6 | 133.1±94.5 | 110.4±61.1 | 116.6±63.9 | 130.4±109.6 | 141.1±86.9 | 153.3±152.6 | <0.001 |
| γ-Glutamyl transferase, mU/mL | 19.3±21.7 | 47.5±86.1 | 28.0±50.2 | 38.5±53.0 | 44.6±64.0 | 84.3±192.8 | 103.7±174.6 | <0.001 |

MET-hours, Metabolic equivalent score; HOMA-IR, homeostasis model assessment of insulin resistance; HOMA-β, homeostasis model assessment of β-cell. Data are means±standard deviations or proportions.

[a] Monthly income <10$^6$ Won.

associations even after controlling for other potential risk factors. Compared with lifetime abstainers, alcohol drinkers had multivariate HRs (95% CI) of 1.74 (1.02, 2.95) for 16–30 g/day, 2.09 (1.16, 3.77) for 31–60 g/day, and 1.94 (1.07, 3.51) for >60 g/day of average alcohol consumption. There was no reduction in the T2DM risk among alcohol drinkers who consumed 15 g/day or less, nor was there among former drinkers, compared with lifetime abstainers

The results of the sex-specific analyses are presented in Table 3.

Among participants who maintained leanness over the follow-up period, compared with lifetime abstainers, increased risks of T2DM were observed in male drinkers consuming ≥16 g/day, and a significant risk was observed among those with alcohol consumption >30 g/day. Among women, former drinkers had a significantly increased risk, whereas drinkers consuming ≥16 g/day did not. Relative to lifetime abstainers, the former drinkers had a multivariate HR (95% CI) of 2.96 (1.02, 8.64) in females. No significant risk reduction was observed in associations with light to moderate alcohol consumption among male and female drinkers.

Table 4 demonstrates the results regarding significant factors associated with T2DM risk obtained from the stepwise analysis for 2014 participants who maintained leanness over the follow-up period.

At a significance level of 0.05, age, parental history of DM, physical activity, and alcohol consumption were finally included in the model. Participants in the top quintile of physical

**Table 2. Association between alcohol consumption and 10-year incidence of type 2 diabetes mellitus.**

| | Hazard ratio (95% confidence interval) | | | | | | |
|---|---|---|---|---|---|---|---|
| | Lifetime | Former | Current drinker: average alcohol consumption (g/day) | | | | |
| | abstainer | drinker | 0.1–5 | 6–15 | 16–30 | 31–60 | >60 |
| All participants with BMI <23 kg/m² at baseline (n = 2338) | | | | | | | |
| Number of subjects (% of cases) | 1138 (9.4) | 149 (15.4) | 382 (10.5) | 255 (10.2) | 175 (15.4) | 130 (13.1) | 109 (18.4) |
| Age and sex-adjusted | reference | 1.45 (0.90, 2.35) | 1.24 (0.85, 1.79) | 1.07 (0.68, 1.70) | 1.66 (1.04, 2.63) | 1.42 (0.82, 2.46) | 1.77 (1.05, 2.96) |
| Multivariate adjusted[a] | reference | 1.50 (0.92, 2.45) | 1.26 (0.87, 1.83) | 1.07 (0.68, 1.71) | 1.63 (1.02, 2.61) | 1.55 (0.89, 2.70) | 1.74 (1.03, 2.94) |
| Participants with BMI <23 kg/m² over 10 years (n = 2014) | | | | | | | |
| Number of subjects (% of cases) | 982 (9.3) | 137 (16.1) | 329 (9.7) | 219 (9.1) | 148 (14.2) | 108 (14.8) | 91 (17.6) |
| Age and sex-adjusted | reference | 1.64 (0.99, 2.74) | 1.19 (0.79, 1.80) | 1.02 (0.61, 1.71) | 1.67 (0.99, 2.82) | 1.87 (1.05, 3.33) | 1.92 (1.08, 3.42) |
| Multivariate adjusted[a] | reference | 1.67 (0.99, 2.79) | 1.22 (0.80, 1.84) | 0.99 (0.58, 1.66) | 1.74 (1.02, 2.95) | 2.09 (1.16, 3.77) | 1.94 (1.07, 3.51) |

BMI, body mass index.

[a] Adjusted for age (continuous), sex, household income (wage <10⁶ Won/month or +10⁶ Won/month), occupation (office worker or non-office worker), educational level (<9 years or +9 years), marital status (married or other), parental history of diabetes mellitus (yes or no), smoking status (never smoked, former smoker, current smoker; <10 cigarettes/day, 11–20 cigarettes/day, +21 cigarettes/day), body mass index (continuous), quintiles of metabolic equivalent score, and quintiles of dietary calorie intake.

activity level had a 46% (95% CI: 19%, 64%) reduction in the risk of T2DM. Those with a parental history of DM, as well as those with alcohol consumption ≥16 g/day, were found to have an approximately two-fold higher risk of developing T2DM compared with their respective reference categories. Regardless of the inclusion of waist circumference in the model, the

**Table 3. Sex-specific association between alcohol consumption and 10-year incidence of type 2 diabetes mellitus among participants who maintained leanness over the follow-up period.**

| | Hazard ratio (95% confidence interval) | | | | |
|---|---|---|---|---|---|
| | Lifetime | Former | Current drinker: average alcohol consumption (g/day) | | |
| | abstainer | drinker | ≤15 | 16–30 | >30 |
| Men with BMI <23 kg/m² over 10 years (n = 1,026) | | | | | |
| Number of participants (% of cases) | 250 (12.0) | 120 (15.0) | 333 (9.6) | 133 (14.3) | 190 (16.8) |
| Age-adjusted | reference | 1.36 (0.76, 2.45) | 0.99 (0.60, 1.62) | 1.47 (0.83, 2.62) | 1.78 (1.08, 2.93) |
| Multivariate adjusted[a] | reference | 1.31 (0.72, 2.38) | 0.92 (0.56, 1.53) | 1.52 (0.84, 2.74) | 1.83 (1.09, 3.06) |
| Women with BMI <23 kg/m² over 10 years (n = 988) | | | | | |
| Number of participants (% of cases) | 732 (8.3) | 17 (23.5) | 215 (9.3) | 24 (8.3)[b] | |
| Age-adjusted | reference | 3.63 (1.30, 10.10) | 1.29 (0.77, 2.15) | 1.46 (0.36, 6.02) | |
| Multivariate adjusted[a] | reference | 2.96 (1.02, 8.64) | 1.32 (0.77, 2.26) | 1.54 (0.36, 6.53) | |

BMI, body mass index.

[a] Adjusted for age (continuous), household income (wage <10⁶ Won/month or +10⁶ Won/month), occupation (office worker or non-office worker), educational level (<9 years or +9 years), marital status (married or other), parental history of diabetes mellitus (yes or no), smoking status (never smoked, former smoker, current smoker; <10 cigarettes/day, 11–20 cigarettes/day, +21 cigarettes/day), body mass index (continuous), quintiles of metabolic equivalent score, and quintiles of dietary calorie intake.

[b] The categories of 16–30 g/day, 31–60 g/day, and > 60 g/day have been combined due to small numbers of cases.

**Table 4. Stepwise results on the association between alcohol consumption and 10-year incidence of type 2 diabetes mellitus in 2,014 participants who maintained leanness over the follow-up period.**

| Variables | | Hazard ratio (95% confidence interval) |
|---|---|:---:|
| Age | | 1.04 (1.02, 1.05) |
| Parental history of diabetes mellitus | | 2.66 (1.81, 3.91) |
| Physical activity | 1st quintile | reference |
| | 2nd quintile | 0.61 (0.38, 0.99) |
| | 3rd quintile | 0.63 (0.41, 0.96) |
| | 4th quintile | 0.70 (0.49, 1.01) |
| | 5th quintile | 0.54 (0.36, 0.81) |
| Alcohol consumption | Lifetime abstainer | reference |
| | Former drinker | 1.87 (1.17, 2.99) |
| | Current drinker with ≤15 g/day | 1.21 (0.86, 1.71) |
| | Current drinker with 16–30 g/day | 1.87 (1.16, 3.01) |
| | Current drinker with >30 g/day | 2.20 (1.46, 3.31) |

Stepwise analysis included age, household income, occupation, educational level, marital status, parental history of diabetes mellitus, smoking status, body mass index, waist circumference, quintiles of metabolic equivalent score, and quintiles of dietary calorie intake.

same results were obtained because waist circumference was not significantly associated with T2DM risk in these participants.

## Discussion

The present study investigated the association of alcohol consumption and potential risk factors with T2DM risk among lean individuals with a BMI ranging from 17 kg/m$^2$ to 22.9 kg/m$^2$. Alcohol consumption ≥2 units/day (≥16 g/day) significantly increased T2DM risk. Additionally, physical activity reduced the T2DM risk independent of age, parental history of DM, and alcohol consumption.

The Korean Diabetes Association recently reported that approximately 14% of Korean adults aged 30 years or older have T2DM and half of them are obese [15]. Thus, the remaining half of Korean adults with T2DM are underweight or have normal body weights, whereas only 12.5% of individuals with T2DM are lean in the US [16]. In terms of ethnicity, Asians are the major contributors and are disproportionately represented among people classified as having lean T2DM in the US [5, 6]. The proportion of individuals with a BMI <25 kg/m$^2$ was 8.4% among German T2DM patients and between 3% and 6% among Caucasian T2DM patients in the US [5, 8]. In contrast, the pooled data for Asian populations included in 22 prospective cohort studies revealed that 65% and 39% of T2DM patients had a BMI <25 kg/m$^2$ and a BMI <23 kg/m$^2$, respectively [17]. Potential risk factors in the causal pathway leading to lean T2DM are still unclear, although some genetic studies have proposed susceptibility genes [17–19], and a few cross-sectional studies have suggested male gender, smoking, and alcohol consumption [6, 8].

The present prospective study attempted to explore risk factors for lean T2DM with a particular focus on alcohol consumption because previous findings regarding its association with T2DM risk have been conflicting [1–3]. Some of the associations differ between the sexes and among different ethnicities [2], and this may reflect sex- and ethnicity-associated obesity patterns, given that obesity is a well-known and strong risk factor for T2DM. Thus, we investigated whether alcohol consumption is a potential risk factor for T2DM among lean

individuals. We observed that current drinkers who consumed ≥2 units/day of alcohol had a significantly higher risk of developing T2DM compared with lifetime abstainers. Former drinkers, notably in females, also had an increased risk of T2DM, suggesting that they should not be included in the reference group as abstainers. If former drinkers were included in the reference group with lifetime abstainers, the current drinkers would have had a falsely reduced risk of T2DM, resulting in a U-shaped association curve [1]. We found that former drinkers had elevated glucose levels and reduced HOMA-β cell levels, indicating impaired glucose tolerance and pancreatic β-cell dysfunction compared with lifetime abstainers. These findings suggest that women might often cease their alcohol consumption at the pre-diabetic stage, probably as part of a conscious effort to modify behavioral factors associated with T2DM and overall health.

In terms of biological mechanisms underlying the association between alcohol consumption and T2DM risk, alcohol produces reactive oxidative species, which impair β-cell mitochondrial function leading to apoptosis [9, 10, 20]. This leads not only to the direct effects of alcohol at the cellular level but also to the consequences of alcohol metabolism, such as acetaldehyde toxicity and excess fatty acid and triglyceride synthesis, which may contribute to the development of T2DM [21].

In this study, we have demonstrated age, parental history of DM, physical activity, and alcohol consumption as significant factors associated with T2DM risk among lean individuals. We estimated a similar effect size for family history of DM as that reported by a previous study, which suggested that family history of DM may reflect genetic susceptibility to DM, adiposity, and other environmental components shared within family members [22]. Genetic susceptibility has been reported to be stronger in lean T2DM than in obese T2DM [18, 23]. Nevertheless, physical activity remained a significant factor in the model independent of age and parental history of DM, and these findings are compatible with previous findings [22]. Thus, regardless of aging and family history of DM, increasing physical activity and abstaining from alcohol may be practical strategies for lean individuals to prevent T2DM. Furthermore, intervention opportunities through health policies and educational programs should be provided to all alcohol drinkers at a high-risk of T2DM.

There are some potential limitations that should be taken into account when interpreting our results. First, information on alcohol consumption was collected via face-to-face, questionnaire-based interviews. This method might make heavy alcohol drinkers underreport alcohol consumption. Second, we observed a lack of association between current alcohol consumption and T2DM risk among women likely because of the small numbers of heavy drinkers. Third, there might be residual confounding in the associations; however, we considered a broad range of potential confounding factors. On the other hand, our study had several strengths, including its prospective design, the use of detailed (in particular, defining lifetime abstainers and former drinkers separately) and updated information on alcohol consumption, and the use of assayed blood glucose levels to define cases. Because the health surveys were administered by trained researchers, the number of non-responders (1.2%) was trivial. The study findings regarding the risk factors for lean T2DM are novel and may be generalizable for Asian populations with low or normal BMIs.

In summary, this prospective study, including middle-aged and older Koreans with low or normal BMIs, observed that any amount of alcohol consumed is not beneficial and alcohol consumption ≥2 units/day significantly increased T2DM risk. Additionally, high physical activity levels decreased T2DM risk, whereas age and parental history of DM increased the risk. On the basis of these findings, abstaining from alcohol and increasing physical activity may be beneficial for the prevention of lean T2DM.

## Author Contributions

**Conceptualization:** Inkyung Baik, Sang Ick Park.

**Data curation:** Inkyung Baik.

**Formal analysis:** Inkyung Baik.

**Funding acquisition:** Inkyung Baik.

**Writing – original draft:** Inkyung Baik.

**Writing – review & editing:** Inkyung Baik, Sang Ick Park.

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
