## [Decision Letter · Decision Letter 0]

29 May 2020

PONE-D-20-03085

Associations of alcohol consumption and physical activity with lean type 2 diabetes mellitus among Korean adults: A prospective cohort study

PLOS ONE

Dear Dr. Baik,

Thank you for submitting your manuscript to PLOS ONE. After careful consideration, we feel that it has merit but does not fully meet PLOS ONE’s publication criteria as it currently stands. Therefore, we invite you to submit a revised version of the manuscript that addresses the points raised during the review process.

The revised version should address all comments in the reports.

We look forward to receiving your revised manuscript.

Kind regards,

Petri Böckerman

Academic Editor

PLOS ONE

Journal Requirements:

"Human Subjects Review Committee of Kookmin University (KMU-201512-HR-094)".

a. Please amend your current ethics statement to confirm that your named institutional review board or ethics committee specifically approved this study.

3. In your Abstract and discussion sections, please ensure to revise statements of causation (such as "To prevent lean T2DM, abstaining from alcohol and physical activity should be encouraged"), as they are not supported by the observational nature of the present study.

4. Please include additional information regarding the survey or questionnaire used in the study and ensure that you have provided sufficient details that others could replicate the analyses.

For instance, if you developed a questionnaire as part of this study and it is not under a copyright more restrictive than CC-BY, please include a copy, in both the original language and English, as Supporting Information.

Moreover, please include more details on how the questionnaire was pre-tested, and whether it was validated.

Reviewers' comments:

Reviewer's Responses to Questions

**Comments to the Author**

1. Is the manuscript technically sound, and do the data support the conclusions?

Reviewer #1: Partly

Reviewer #2: Yes

2. Has the statistical analysis been performed appropriately and rigorously? 

Reviewer #1: Yes

Reviewer #2: Yes

3. Have the authors made all data underlying the findings in their manuscript fully available?

Reviewer #1: Yes

Reviewer #2: Yes

4. Is the manuscript presented in an intelligible fashion and written in standard English?

Reviewer #1: Yes

Reviewer #2: Yes

5. Review Comments to the Author

Reviewer #1: Referee report on "Associations of alcohol consumption and physical activity with lean type 2 diabetes mellitus among Korean adults: A prospective cohort study" (PONE-D-20-03085)

In this paper, the authors examined associations between alcohol consumption and the risk of type 2 diabetes mellitus (T2DM) among lean Korean adults. Based on the results, alcohol drinkers who consumed alcohol at least 16g/day were more likely to develop T2DM over a 10 -year period than life-time abstainers. Physical activity decreased the risk of T2DM. Obesity is one of the leading risk factors of T2DM but risk factors among lean individuals have received less attention. The article is very clear and pleasant to read. However, I wish there was a bit more elaboration on the context. I am also somewhat concerned about the number of observations which seems to be low in some regressions. Below are my comments on this paper.

1. In the introduction, it should be explicitly stated, why it is important to investigate the incidence of T2DM among lean individuals.

2. Why alcohol consumption may affect the incidence of T2DM? On pages 19-20 the authors provide one potential explanation but I suppose there are also other explanations? I would like to see discussion of why alcohol consumption may affect the incidence of T2DM already in the introduction.

3. Not only the volume but also patterns of drinking may have implications on the association between alcohol consumption and health. For example, the type of alcohol (e.g. wine vs. spirits) and the way alcohol is consumed (e.g. with meals vs. binge drinking) may have implications on the association between alcohol consumption and T2DM. The authors have access to detailed information on alcohol consumption. They could utilize their data to explore whether different patterns of drinking have implications on their results or at least discuss the potential role of different drinking patterns.

4. It looks like the number of individuals (number of cases) does not sum up to n = 2338 in table 2 and the same problem applies also to Table 3. Also, if I read the tables correctly, the number of observations in some groups seems to be very low (< 20 in Table 2 and < 10 in Table 3) which makes it very challenging to detect reasonable-size effects with reasonable power.

5. The authors note that including both lifetime abstainers and former drinkers to the reference group would lead to biased conclusions. This finding has been reported also elsewhere (see e.g. Baliunas et al. 2009) and the authors could refer to these studies as well.

6. Data description: Page 4, line 69: “A population-based prospective study included male and female Koreans aged 40 to 69 years”. Please clarify, was this the age at the baseline (if not, when)? Also, please clarify whether information on BMI (and T2DM) were based on self-reported information or medical examinations.

References

Baliunas, D. O., Taylor, B. J., Irving, H., Roerecke, M., Patra, J., Mohapatra, S., & Rehm, J. (2009). Alcohol as a risk factor for type 2 diabetes: a systematic review and meta-analysis. Diabetes care, 32(11), 2123-2132.

Reviewer #2: Comments

1. The exact contribution to the (international) literature should be stated in the revised introduction.

2. What is the motivation for the focus on Korea?

3. The paper uses information on health surveys. Was non-response to the surveys random or not? This information would be useful in order to better understand the estimation results. If those with least physical activity are less likely to respond to the health surveys, the estimates that are obtained may be biased, at least to some degree. This issue should be noted in the paper.

4. Do the data contain (survey) weights or not? Why they have not been used in the estimations?

5. Do the models contain the relevant controls? Physical activity is linked to education (https://doi.org/10.1111/sms.13653). This issue should be noted in the revised version.

6. The paper does not consider the potential heterogeneity in the estimated effects. The relationships can differ significantly e.g. by gender.

7. The concluding section of the paper should discuss more about the practical policy lessons that can be drawn from the estimation results.

6. PLOS authors have the option to publish the peer review history of their article (what does this mean?). If published, this will include your full peer review and any attached files.

Reviewer #1: No

Reviewer #2: No

---

## [Author Response · Author response to Decision Letter 0]

2 Jul 2020

Re: PONE-D-20-03085

Associations of alcohol consumption and physical activity with lean type 2 diabetes mellitus among Korean adults: A prospective cohort study

Here are our responses to the comments for the above-referenced manuscript.

Response to the Reviewer #1’s comments: 

1. In the introduction, it should be explicitly stated, why it is important to investigate the incidence of T2DM among lean individuals.

Response: Thank you for your valuable comments. We revised the Introduction part according to the comment (lines 60-68, revised).

2. Why alcohol consumption may affect the incidence of T2DM? On pages 19-20 the authors provide one potential explanation but I suppose there are also other explanations? I would like to see discussion of why alcohol consumption may affect the incidence of T2DM already in the introduction.

Response: We agree on this opinion. We revised the Introduction part according to the comment (lines 60-68, revised).

3. Not only the volume but also patterns of drinking may have implications on the association between alcohol consumption and health. For example, the type of alcohol (e.g. wine vs. spirits) and the way alcohol is consumed (e.g. with meals vs. binge drinking) may have implications on the association between alcohol consumption and T2DM. The authors have access to detailed information on alcohol consumption. They could utilize their data to explore whether different patterns of drinking have implications on their results or at least discuss the potential role of different drinking patterns.

Response: In terms of alcoholic beverages, almost all alcohol drinkers consumed ‘soju’, which is a major type of alcoholic beverage in Korea and similar to ‘sake’. So, we were unable to conduct separate analyses according to the types of alcoholic beverage. The information on the consumption of meals or specific foods in the occasion of alcohol drinking was unavailable. Among alcohol drinkers who consumed 30g/day or greater, about 20% were binge drinkers. We found no significant association between binge drinking and T2DM partly due to a small number of binge drinkers. We now added information on the types of alcoholic beverage (lines 166-168, revised).

4. It looks like the number of individuals (number of cases) does not sum up to n = 2338 in table 2 and the same problem applies also to Table 3. Also, if I read the tables correctly, the number of observations in some groups seems to be very low (< 20 in Table 2 and < 10 in Table 3) which makes it very challenging to detect reasonable-size effects with reasonable power.

Response: We have now corrected the number of participants (Tables 2 and 3, revised).

5. The authors note that including both lifetime abstainers and former drinkers to the reference group would lead to biased conclusions. This finding has been reported also elsewhere (see e.g. Baliunas et al. 2009) and the authors could refer to these studies as well.

Response: Thank you for this comment. The original list of references included the study of Baliunas et al. (2009). We added the reference number of this article in the Discussion part (line 260, revised). 

6. Data description: Page 4, line 69: “A population-based prospective study included male and female Koreans aged 40 to 69 years”. Please clarify, was this the age at the baseline (if not, when)? Also, please clarify whether information on BMI (and T2DM) were based on self-reported information or medical examinations.

Response: We have now clarified the description regarding age, BMI, and T2DM according to the comment (lines 110-111 and 137-138, revised).

Response to the Reviewer #2’s comments: 

1. The exact contribution to the (international) literature should be stated in the revised introduction.

Response: Thank you for your valuable comments. We re-checked out all references quoted in the Introduction part. The reference 7 was added (line 58) because we estimated the prevalence of lean T2DM in a Korean population using the Korean NHANES data. There was no previous literature regarding such an estimate.

2. What is the motivation for the focus on Korea?

Response: We added the text regarding the motivation (lines 64-68, revised).

3. The paper uses information on health surveys. Was non-response to the surveys random or not? This information would be useful in order to better understand the estimation results. If those with least physical activity are less likely to respond to the health surveys, the estimates that are obtained may be biased, at least to some degree. This issue should be noted in the paper.

Response: Because the health surveys were administered by trained researchers, the number of non-responders to the surveys including specific questions on physical activity was trivial. Thus, we think that the effects of non-responders on the study findings were insignificant. We have now noted this aspect in the manuscript (lines 294-295, revised). 

4. Do the data contain (survey) weights or not? Why they have not been used in the estimations?

Response: The present study used data from a whole cohort and applied the inclusion and exclusion criteria. If the study used data from a sample which was selected from an original cohort, survey weights might have been considered in the analysis.

5. Do the models contain the relevant controls? Physical activity is linked to education (https://doi.org/10.1111/sms.13653). This issue should be noted in the revised version.

Response: The models included educational level with other potential confounding factors. This issue has been noted in the line 290.

6. The paper does not consider the potential heterogeneity in the estimated effects. The relationships can differ significantly e.g. by gender.

Response: We agree on this comment. Because we also thought that the associations can differ by gender, we presented data stratified by gender in Table 3.

7. The concluding section of the paper should discuss more about the practical policy lessons that can be drawn from the estimation results.

Response: We have now added a text according to the comment (lines 282-283, revised).

Response to the editor’s comments:

Response: We checked it out.

2. Please amend your current ethics statement to confirm that your named institutional review board or ethics committee specifically approved this study.

Response: We have now revised the ethics statement (lines 97-105, revised).

3. In your Abstract and discussion sections, please ensure to revise statements of causation (such as "To prevent lean T2DM, abstaining from alcohol and physical activity should be encouraged"), as they are not supported by the observational nature of the present study.

Response: We have now revised it (lines 35-36, revised).

4. Please include additional information regarding the survey or questionnaire used in the study and ensure that you have provided sufficient details that others could replicate the analyses. For instance, if you developed a questionnaire as part of this study and it is not under a copyright more restrictive than CC-BY, please include a copy, in both the original language and English, as Supporting Information. Moreover, please include more details on how the questionnaire was pre-tested, and whether it was validated.

Response: Because of the copyright that The Korea Centers for Disease Control & Prevention owns, we are unable to provide a questionnaire copy. In terms of the validity of alcohol consumption, additional information has been now added (lines 124-127, revised).

---

## [Decision Letter · Decision Letter 1]

20 Jul 2020

PONE-D-20-03085R1

Associations of alcohol consumption and physical activity with lean type 2 diabetes mellitus among Korean adults: A prospective cohort study

PLOS ONE

Dear Dr. Baik,

Thank you for submitting your manuscript to PLOS ONE. After careful consideration, we feel that it has merit but does not fully meet PLOS ONE’s publication criteria as it currently stands. Therefore, we invite you to submit a revised version of the manuscript that addresses the points raised during the review process.

The revised version should address the remaining comments.

We look forward to receiving your revised manuscript.

Kind regards,

Petri Böckerman

Academic Editor

PLOS ONE

Reviewers' comments:

Reviewer's Responses to Questions

**Comments to the Author**

1. If the authors have adequately addressed your comments raised in a previous round of review and you feel that this manuscript is now acceptable for publication, you may indicate that here to bypass the “Comments to the Author” section, enter your conflict of interest statement in the “Confidential to Editor” section, and submit your "Accept" recommendation.

Reviewer #1: (No Response)

Reviewer #2: All comments have been addressed

2. Is the manuscript technically sound, and do the data support the conclusions?

Reviewer #1: Yes

Reviewer #2: Yes

3. Has the statistical analysis been performed appropriately and rigorously? 

Reviewer #1: Yes

Reviewer #2: Yes

4. Have the authors made all data underlying the findings in their manuscript fully available?

Reviewer #1: Yes

Reviewer #2: No

5. Is the manuscript presented in an intelligible fashion and written in standard English?

Reviewer #1: Yes

Reviewer #2: Yes

6. Review Comments to the Author

Reviewer #1: I think the paper has improved. However, there are still two issues the authors should consider.

1. I asked the authors to explain in the introduction, why it is important to investigate the incidence of T2DM among lean individuals. The only explanation they give is that there is not much previous evidence on this topic. However, I think just the fact that there is not much evidence does not justify the topic. So, why is it important to investigate the incidence of T2DM among lean individuals?

2. Reviewer 2 brought up an excellent point about non-response. In the revised version, the authors state that “Because the health surveys were administrated by trained researchers, the number of non-responders was trivial”. As long as a survey is voluntary, there is a significant risk that non-random non-response occurs. The authors should provide exact information concerning non-response.

Reviewer #2: I am happy with the revised version of the paper. I like the research question, the structure of the paper, the quality of writing, and the way the authors describe their empirical proceeding and results. Most importantly, the authors have addressed all the issues stated in my referee report for the first version appropriately.

7. PLOS authors have the option to publish the peer review history of their article (what does this mean?). If published, this will include your full peer review and any attached files.

Reviewer #1: No

Reviewer #2: No

---

## [Author Response · Author response to Decision Letter 1]

5 Aug 2020

Re: PONE-D-20-03085

Associations of alcohol consumption and physical activity with lean type 2 diabetes mellitus among Korean adults: A prospective cohort study

Here are our responses to the comments for the above-referenced manuscript.

Response to the Reviewer #1’s comments: 

Reviewer #1: 

1. I asked the authors to explain in the introduction, why it is important to investigate the incidence of T2DM among lean individuals. The only explanation they give is that there is not much previous evidence on this topic. However, I think just the fact that there is not much evidence does not justify the topic. So, why is it important to investigate the incidence of T2DM among lean individuals?

Response: Thank you for further comments. We have now added contents related to the importance of research for lean T2DM (lines 58-60 and 70-72, revised).

2. Reviewer 2 brought up an excellent point about non-response. In the revised version, the authors state that “Because the health surveys were administrated by trained researchers, the number of non-responders was trivial”. As long as a survey is voluntary, there is a significant risk that non-random non-response occurs. The authors should provide exact information concerning non-response.

Response: We have now added the exact proportion of non-responders (1.2%) (line 299, revised).

Reviewer #2: I am happy with the revised version of the paper. I like the research question, the structure of the paper, the quality of writing, and the way the authors describe their empirical proceeding and results. Most importantly, the authors have addressed all the issues stated in my referee report for the first version appropriately.

Response: Thank you very much for your valuable comments.

---

## [Decision Letter · Decision Letter 2]

21 Aug 2020

Associations of alcohol consumption and physical activity with lean type 2 diabetes mellitus among Korean adults: A prospective cohort study

PONE-D-20-03085R2

Dear Dr. Baik,

We’re pleased to inform you that your manuscript has been judged scientifically suitable for publication and will be formally accepted for publication once it meets all outstanding technical requirements.

Kind regards,

Petri Böckerman

Academic Editor

PLOS ONE

Additional Editor Comments (optional):

Reviewers' comments:

Reviewer's Responses to Questions

**Comments to the Author**

1. If the authors have adequately addressed your comments raised in a previous round of review and you feel that this manuscript is now acceptable for publication, you may indicate that here to bypass the “Comments to the Author” section, enter your conflict of interest statement in the “Confidential to Editor” section, and submit your "Accept" recommendation.

Reviewer #1: All comments have been addressed

Reviewer #2: All comments have been addressed

2. Is the manuscript technically sound, and do the data support the conclusions?

Reviewer #1: Yes

Reviewer #2: Yes

3. Has the statistical analysis been performed appropriately and rigorously? 

Reviewer #1: Yes

Reviewer #2: Yes

4. Have the authors made all data underlying the findings in their manuscript fully available?

Reviewer #1: Yes

Reviewer #2: No

5. Is the manuscript presented in an intelligible fashion and written in standard English?

Reviewer #1: Yes

Reviewer #2: Yes

6. Review Comments to the Author

Reviewer #1: (No Response)

Reviewer #2: I am happy with the revised version of the paper. I like the research question, the structure of the paper, the quality of writing, and the way the authors describe their empirical proceeding and results. Most importantly, the authors have addressed all the issues stated in my referee report for the first version appropriately.

7. PLOS authors have the option to publish the peer review history of their article (what does this mean?). If published, this will include your full peer review and any attached files.

Reviewer #1: No

Reviewer #2: No

---

## [Editor Report · Acceptance letter]

26 Aug 2020

PONE-D-20-03085R2 

Associations of alcohol consumption and physical activity with lean type 2 diabetes mellitus among Korean adults: A prospective cohort study 

Dear Dr. Baik:

I'm pleased to inform you that your manuscript has been deemed suitable for publication in PLOS ONE. Congratulations! Your manuscript is now with our production department. 

Kind regards, 

on behalf of

Professor Petri Böckerman 

Academic Editor

PLOS ONE